# Neurofeedback for the Education of Children with ADHD and Specific Learning Disorders: A Review

**DOI:** 10.3390/brainsci12091238

**Published:** 2022-09-14

**Authors:** Abhishek Uday Patil, Deepa Madathil, Yang-Tang Fan, Ovid J. L. Tzeng, Chih-Mao Huang, Hsu-Wen Huang

**Affiliations:** 1Department of Biological Science and Technology, National Yang Ming Chiao Tung University, Hsinchu 300093, Taiwan; 2Jindal Institute of Behavioural Sciences, O.P. Jindal Global University, Haryana 131001, India; 3Graduate Institute of Medicine, Yuan Ze University, Taoyuan 320315, Taiwan; 4Centre for Intelligent Drug Systems and Smart Bio-Devices (IDS^2^B), National Yang Ming Chiao Tung University, Hsinchu 300093, Taiwan; 5College of Humanities and Social Sciences, Taipei Medical University, Taipei 106339, Taiwan; 6Department of Educational Psychology and Counseling, National Taiwan Normal University, Taipei 106308, Taiwan; 7Hong Kong Institute for Advanced Studies, City University of Hong Kong, Hong Kong; 8Department of Linguistics and Translation, City University of Hong Kong, Hong Kong

**Keywords:** neurofeedback, EEG, ADHD, dyslexia, education

## Abstract

Neurofeedback (NF) is a type of biofeedback in which an individual’s brain activity is measured and presented to them to support self-regulation of ongoing brain oscillations and achieve specific behavioral and neurophysiological outcomes. NF training induces changes in neurophysiological circuits that are associated with behavioral changes. Recent evidence suggests that the NF technique can be used to train electrical brain activity and facilitate learning among children with learning disorders. Toward this aim, this review first presents a generalized model for NF systems, and then studies involving NF training for children with disorders such as dyslexia, attention-deficit/hyperactivity disorder (ADHD), and other specific learning disorders such as dyscalculia and dysgraphia are reviewed. The discussion elaborates on the potential for translational applications of NF in educational and learning settings with details. This review also addresses some issues concerning the role of NF in education, and it concludes with some solutions and future directions. In order to provide the best learning environment for children with ADHD and other learning disorders, it is critical to better understand the role of NF in educational settings. The review provides the potential challenges of the current systems to aid in highlighting the issues undermining the efficacy of current systems and identifying solutions to address them. The review focuses on the use of NF technology in education for the development of adaptive teaching methods and the best learning environment for children with learning disabilities.

## 1. Introduction

Neurofeedback (NF) is a type of biofeedback, the most common and traditional form of which is the electroencephalogram (EEG) NF (EEG-NF). EEG-NF displays EEG waves that are measured via electrodes placed on the scalp. The brain is constantly active, whether one is awake or asleep and whether in the presence or absence of distinct stimuli. EEG waves reflect the brain’s current functional state and can be characterized and separated into delta, theta, alpha, sensorimotor rhythm (SMR), beta, and gamma frequency bands. Each band is a unique indicator of brain activity. For example, brain activity during a complex cognitive task varies from that at rest, and the strengths of the EEG bands vary correspondingly. EEG-NF training captures characteristic EEG bands in real time and also measures the changes in EEG activity that are critical for and promote changes in a targeted cognitive function. In recent years there has been a significant increase in clinical applications of the technique for several neuropsychiatric conditions, including attention-deficit/hyperactivity disorder (ADHD), epilepsy, migraine, and headaches [1,2,3,4,5].

NF methods are usually based on EEG, functional magnetic resonance imaging (fMRI), or near-infrared spectroscopy (NIRS). Hitherto, EEG has been the primary method used in NF, mainly because of its low cost, ease of use, and non-invasiveness [6]. Technological advances have now made it possible to combine methods, as in the simultaneous EEG-fMRI method in which the subject wears an EEG electrode cap while inside an MRI scanner. In this advanced method, EEG and fMRI recordings are performed simultaneously [7]. Advances in fMRI-NF have demonstrated promising results with techniques such as decoded NF (DecNef) and functional connectivity NF (FCNef) [8,9,10,11]. In the current report, we restrict our focus to EEG-NF.

EEG comprises a mixture of multiple waves of various frequencies. These frequency bands are correlated with various degrees of neuronal synchronizations [12] and are also linked to a variety of cognitive operations (Harmony, 2013). For example, delta waves (0.5–4 Hz) are associated with inhibition of the sensory afferents [13,14]; theta waves (4–8 Hz) are typical of nervousness [15,16]; an attentive and relaxed state of mind is characterized by alpha waves (8–12 Hz) [17,18,19]; alertness is characterized by beta waves (16–30 Hz) [20]; and problem-solving or higher cognitive functions are associated with gamma waves (30–100 Hz) [21,22]. In terms of resting-state EEG frequency bands, waking state eyes-open activity is correlated with beta waves, whereas stage I non-rapid eye movement (non-REM) sleep is characterized by theta waves. Stage II non-REM sleep is characterized by spindles in the 10–15 Hz frequency range, whereas stage III and IV non-REM sleep are characterized by slow delta waves. Rapid eye movement (REM) sleep, after this stage, is characterized by high frequency and low amplitude waves, which is similar to the brain activity when individuals are awake.

Some characteristics of the EEG components and their frequency ranges are presented in Table 1. EEG-NF protocols usually depend on the power of various EEG frequencies such as the delta, theta, alpha, and beta bands or the alpha/theta or beta/theta ratios [23].

NF systems consist of five basic components, as illustrated in Figure 1. The first is the data acquisition component, which is used to acquire brain signals. Neural responses can be recorded using EEG, electrocorticography (ECoG), intracranial EEG, or magnetoencephalography (MEG). The advantage of using these methods for NF is their high temporal resolution, which implies that brain activities are continuously and directly reflected within milliseconds. High spatial resolution fMRI and near-infrared spectroscopy (NIRS) are also being increasingly used for data acquisition. The second component is online data processing, which involves the detection and removal of artifacts from the signals provided by the data acquisition component. Typical artifacts from EEG/ECoG/MEG include muscle or eye movements, and 50/60 Hz power line interference. Further to this, artifacts associated with EEG at the scalp are a mix of a spatial smearing effect from the skull and other tissues and multiple sources of brain activity [24]. Placing electrodes close to each other nearby reduces the number of artifacts and results in a better signal from the scalp. The blind source separation (BSS) method is typically used to separate artifacts from signals such as the EEG [25,26,27]. This method detects signal sources that are important for understanding brain activity [28]. In fMRI-based NF, participants attempt to control their brain activity using real-time feedback from a particular region or network [29,30]. The most common artifacts in the NIRS/fMRI signal are head motion, body motion, respiration, and heartbeat. Artifacts caused by inhomogeneity in the magnetic field are also common in fMRI-NF. Similar to the case of EEG, good signal processing methods such as BSS can separate these artifacts from fMRI signals [31].

The third component is online feature extraction, which involves extracting the brain activity in time or frequency domains from the artifact-free signals. Studies suggest that insubstantial feature extraction can cause an NF system to fail [32]. Currently, machine learning and sophisticated very large-scale integrated chips suitable for online feature extraction are not used in NF systems. Instead, the fast Fourier transform (FFT) and wavelet transform (WT), which provide information about the frequency domain and time domain, respectively, are typically used for feature extraction. Recent technical developments have also made it possible to analyze data online for fMRI- and NIRS-based NF systems [30,31].

The fourth component of NF systems is continuous audio/video feedback, which presents the brain activity to the participant while they perform a task. The feedback can be audio, visual, or a combination, and is intended to help participants modulate their brain activity to complete the task and obtain a reward. Specifically, it shows brain activity about a task and presents a reward when the feature crosses a particular threshold, for instance, >75% theta. The final component of NF systems is the participant’s performance, which determines the success or failure of the NF training. The difficulty of the task can be altered according to how easily the participant can perform the task.

NF learning is based on a control-theoretical framework with a closed-loop pipeline. In practice, the initial stage of NF is characterized by fluctuating feedback, which reflects unconditioned neural variability [33]. This neural variability may then be tuned so that the brain activity reaches the threshold for rewarding feedback during the training. Ideally, brain oscillations that afford an optimal balance between network flexibility and stability can be achieved after learning. This would involve a series of learning events from a neurophysiological mechanism, such as the activation of dopamine receptors. Although a few studies have shed light on how neurotransmitter circuits modulate brain oscillations and hence support brain plasticity [33,34], as yet there is no encompassing theory on this topic.

NF learning is a promising treatment option in clinical practice. Studies have shown that NF has positive effects on multiple domains of clinical outcomes in individuals with neurological (i.e., epilepsy [35]), psychiatric (i.e., depression [36]; drug addiction and alcoholism [37]; schizophrenia [38,39,40]; insomnia [41]), neurodevelopmental (i.e., ADHD [2,42,43]; ASD [44]) and learning disorders (i.e., dyslexia [45]; and specific learning disorders [46,47]). In addition to its medical applications, NF has also been used for performance improvement in sports [48], arts [49], and memory [17], and in the amelioration of confusion, anxiety, and distraction [50,51]. However, despite the multiple promising case reports on NF, there is a lack of reliable empirical studies with consistent results, thus making its clinical use controversial. Nevertheless, other neuroimaging methods such as fMRI have expanded the scope of NF in terms of increased training efficacy in specific clinical domains [52,53,54].

The review discusses the potential for translational application of NF and how it could be useful in education. In this review article, we first present a summary of a generalized NF system and then discuss the use of NF in the treatment of dyslexia, ADHD, and other specific learning disorders. Then, we make recommendations for further research and potential applications of NF in educational settings. NF-based techniques are used to treat dyslexia, ADHD, and other particular learning disorders including dysgraphia and dyscalculia. To assess the potential of NF for educational settings, the review suggests first understanding the current type of feedback provided to children, as well as the potential, challenges, and limitations of the current systems, and provides future directions to aid in identifying the issues undermining the efficacy of current systems and identifying solutions to address them.

## 2. Methods

The literature search was conducted using the PubMed database for peer-reviewed EEG-NF studies on dyslexia, ADHD, and other specific learning disorders up to 31 May 2021. EEG-NF studies on these disorders were included in this review. The search terms used in this review were: (“Dyslexia” OR “reading disability”) AND (“EEG” OR “Electroencephalography”) AND “Neurofeedback” for dyslexia, (“ADHD” OR “ADD” OR “Attention Deficit Hyperactivity Disorder” OR “Attention Deficit Disorder”) AND (“EEG” OR “Electroencephalography”) AND “Neurofeedback for ADHD” and “Learning disorder” AND (“EEG” OR “Electroencephalography”) AND “Neurofeedback” for learning disorders. The search yielded 198 studies, and 178 studies remained after removing duplicates.

The following criteria were adopted for the inclusion of candidate studies (see Figure 2). (1) Studies that were empirical and used EEG-NF with clinical or randomized control trials. (2) Participants included in the study were less than 18 years old. (3) The articles reporting the studies were available in English. Studies that used fMRI, NIRS, and other modalities of NF were excluded from the review.

Twenty-one studies met these criteria and were included in this review. Two NF studies for training children with dyslexia, 14 NF studies for training children with ADHD, and 5 NF studies for training children with other specific learning disorders such as dysgraphia and dyscalculia are included in the review (refer to Tables 2–4).

## 3. Results

The 21 articles were divided into three categories: dyslexia, ADHD, and other specific learning disorders. Table 2 summarizes the research on NF for children with dyslexia, Table 3 the research on NF for children with ADHD symptoms, and Table 4 the research on NF for children with dysgraphia, dyscalculia, and other specific learning disorders.

### 3.1. EEG–Neurofeedback for Assisting Children with Dyslexia

Dyslexia, also called reading disability, is a neurodevelopmental disorder in which children experience heightened difficulty in learning. This difficulty is characterized by inaccurate or slow word recognition, poor spelling ability, and difficulty in decoding. A major challenge for children with dyslexia is translating written words into sounds, which is referred to as a phonological deficit. Children with dyslexia also sometimes have comorbid difficulties in reading comprehension or mathematics comprehension. Dyslexia is one of the most common developmental problems seen in children, with prevalence rates ranging from 5% to 10% in Western societies [55]. A large body of evidence on the cognitive processes associated with dyslexia directly relates to deficits in phonological processing such as phonological awareness, phenome representation, storage, and recall. Neurobiological evidence from children with dyslexia indicates reduced activation in the left hemisphere, including the temporoparietal regions [56]. A recent meta-analysis of fMRI, MEG, and PET studies showed that children with dyslexia exhibit differences in functional connectivity between the left inferior frontal gyrus, supplementary motor area, and middle frontal gyrus compared with typical readers [57]. The disruption of the left temporoparietal regions is related to difficulties in phonological processing and is associated with the under-development of white matter fibers in this region [58]. EEG studies have provided additional insights into the differences in the oscillatory brain activities between children with and without dyslexia during the resting state. Children with dyslexia have demonstrated decreased alpha and beta power bilaterally but increased theta power in the left hemisphere [59].

A few studies have examined the utility of NF for assisting children with dyslexia. Breteler et al. [45] studied the effect of NF sessions on reading and spelling ability in 19 children diagnosed with dyslexia. Personalized NF training protocols were developed based on a quantitative EEG (qEEG) assessment. The children’s spelling ability was tested before and after 40 NF sessions. Power was measured in the frontotemporal region and coherence was measured in the frontocentral and parietal regions. There was a significant increase in alpha coherence and improvement in spelling ability following the NF training. This finding suggests that NF may contribute to the treatment of dyslexia through attention modulation. In another study involving children with dyslexia (N = 6; mean age = 9 years) by Nazari et al. [60], the NF training protocol was designed to reduce delta (1–4 Hz) and theta (4–8 Hz) at T3 (mid-temporal electrode) and enhance beta (15–18 Hz) at F7 (inferior frontal electrode) [61]. Twenty NF sessions, each lasting 30 min, were conducted and interactive video games were used as feedback. The children demonstrated improvements in reading ability following the NF training sessions, evidenced by reductions in reading time and reading mistakes. The evaluations were then repeated after 2 months. The results of the follow-up demonstrated changes in the EEG band power and normalization of coherence in the delta, theta, and beta bands, and concurrent improvements in reading ability and phonological awareness. The authors suggested that these neurophysiological changes in coherence may indicate the integration of sensory and motor areas, which could explain the improvements in reading skills and phonological awareness.

Although there is a paucity of studies on the utility of NF for assisting children with dyslexia, it is supported by substantial anecdotal evidence. For example, cognitive devices can be developed for children with dyslexia to improve their reading time and reduce errors and spelling mistakes based on the studies presented in this review. The NF task must be motivating [62] so that the learner is attentive, ignores distractions, and develops skills related to memory and tasks. The development of such devices would result in a further understanding of the neural correlates of dyslexia based on EEG.

**Table 2 brainsci-12-01238-t002:** Summary of studies on dyslexia and neurofeedback.

Author(s)	Characteristics of the Sample	Groups	NF Details	Post-NF		
				Brain Activities	Behavior (Short Term)	Behavior (Long Term)
Breteler et. al. [45]	n = 19 (11 male) with dyslexia were randomized into 2 groups; age range = 8–16 years, mean age = 10.33 years.	Exp. = 10, control = 9.	1. Increased activity in delta with Z >1.5 times normal at T6. 2. Increased coherence in alpha/beta band at F7–FC3 or F7–C3 with Z > 1.5. 3. Increased coherence at T3–T4 with Z > 1.5.	Increased coherence in delta and alpha bands, and decreased coherence in the beta band.	Improvement in spelling but no clear improvement in reading abilities.	There was a significant improvement in spelling due to NF training. Attentional modulation can be assumed to be involved in this improvement.
Nazari et. al. [60]	n = 6 (all male) with dyslexia; age range = 8–10 years, mean age = 9 years.	Exp. = 6, no control group.	EEG was converted to specific frequency bands using FFT (delta: 1–4 Hz, theta: 4–8 Hz, alpha: 8–12 Hz; beta: 12–25 Hz); z-scores of coherence, absolute power, and relative power were calculated for all the above bands; the study examined individual differences in subjects.	Increased coherence in beta, theta, and delta band. No change in power for all bands.	Reduction in reading errors and reading time following NF sessions.	Sensory–motor integration and cerebral maturity in children with dyslexia.

Note: Exp: experimental group; NFT: neurofeedback training.

### 3.2. EEG–Neurofeedback for Assisting Children with ADHD Symptoms

ADHD is a neurodevelopmental disorder that is most prevalent among children and is categorized by its behavioral presentation [63]. Typical ADHD symptoms include inattentiveness, impulsive behavior, and restlessness, with periods of increased frequency and severity, and the problems persist from childhood to adulthood. About 3–8% of children worldwide have ADHD. It is considered extremely comorbid and is associated with various psychiatric disorders. The prognosis of comorbid ADHD in children is more challenging than that of ADHD only, and the comorbidities change with time and developmental changes. Symptoms of language disorder and oppositional defiant disorder are found in early childhood, whereas tics and anxiety are found in the mid-schooling years. In addition to these, many children with ADHD have a specific learning disorder. Thus, ADHD and its comorbidities pose both clinical and diagnostic challenges. Although ADHD is not considered a learning disability, it does make learning difficult. Importantly, many students with ADHD also have comorbid reading and language disabilities in addition to the deficits directly associated with ADHD. The late development of fronto-cerebellar networks in those with ADHD results in deficits in higher-level executive cognitive functions [64]. Consequently, the condition is associated with poor social and academic outcomes [65]. From a neurobiological perspective, ADHD deficits are most prominent in subcortical areas such as the insula and basal ganglia [66]. Meta-analyses of studies on ADHD have found volume reductions in these subcortical regions as well as in the hippocampus and amygdala. Studies have also found issues in the ventromedial frontal regions [66]. The cortical thickness has been shown to have delayed maturation in the temporal, parietal, and frontal regions in children with ADHD [67].

The clinical use of NF for ADHD is fairly common. The EEG frequency bands of interest are the theta, beta, and alpha bands, and comparisons such as the theta/beta power and amplitude ratios are also useful. Studies have suggested that beta waves represent concentration and increased anxiety, whereas slow theta activity represents thinking without concentration [68], distraction, dreaming, and sleepiness [69]. The SMR frequency band (12–16 Hz) is associated with silent motor activity, alertness, and a calm attentive state [70]. Lubar et al. [71] suggested using the theta/beta ratio as a biomarker to differentiate between children with and without specific learning disorders, attention-deficit disorder (ADD), or ADHD symptoms. The ratio was later used in many other studies such as that of [72], which found that inattention in children with ADHD symptoms was characterized by an increased theta/beta ratio. Specifically, based on two conditions (eyes-closed and eyes-open), the study found that the theta/beta power ratios at Cz were useful for the diagnosis of ADHD [72]. Findings have also suggested that children with ADHD symptoms show consistent activity in the frontocentral regions compared with children without ADHD symptoms [73]. Furthermore, reducing the theta/beta ratio can regulate brain activity associated with attention [42]. That study found a large effect size (0.99) for inattention and a medium effect size (0.55) for hyperactivity.

There are three preferred NF protocols [74] for clinical use in children with ADHD. The first is the suppression of theta activity and enhancement of beta activity. This helps in reducing inattention and impulsivity [74,75]. Table 3 summarizes the studies that have examined this protocol. The objective of the theta/beta ratio protocol is to inhibit theta waves and enhance beta waves. Kropotov et al. [34] used beta and SMR training and observed improvements in children with ADHD symptoms (*n* = 86; mean age = 11.4 years) based on a go/no-go task. The training included beta training at C3-Fz and SMR training at C4-Pz over a range of 15–22 sessions. The study further found a 25% increase in the beta and SMR power at the end of the training compared with during the first session. Doren et al. [76] performed theta/beta NF training to improve the reading ability of children with ADHD (*n* = 31; age = 10–15 years old). A short-term theta/beta protocol at Cz (the midline central electrode) was used in the study. The children with ADHD were able to reduce the theta/beta ratio, which was associated with an improvement in their reading ability. A study by Janssen et al. [77] found a linear decrease in the theta/beta index in children with ADHD symptoms (*n* = 38; mean age = 9.87 years) over 29 sessions. Learning improved over the sessions, which was indicated by the increase in beta power. Another study by Bakhshayesh et al. [75] compared EMG biofeedback with EEG-NF in children with ADHD symptoms (*n* = 35; mean age = 9.34 years). The study found a decreased theta/beta ratio in the NF group, which correlated with an improvement in ADHD symptoms over 30 sessions. It found medium effect sizes for impulsivity and hyperactivity and a large effect size for inattention. Escolano et al. [78] studied the effect of increasing upper alpha power in children with ADHD symptoms (*n* = 20; mean age = 11.8 years). They focused on increasing upper alpha frequencies averaged over the frontal and central sites using an individual alpha frequency and found that the power enhancement of the upper alpha band helped children with ADHD symptoms. The study found large effect sizes for hyperactivity and impulsivity.
brainsci-12-01238-t003_Table 3Table 3Summary of studies on ADHD and neurofeedback.Author(s)Characteristics of the SampleGroupsNF DetailsPost-NFBrain ActivitiesBehavior (Short Term)Behavior (Long Term)Kropotov et al., 2005 [34]*n* = 86 (9 female) with ADHD;age range = 9–14 years,mean age = 11.4 years.Exp. = 86,no control group.1. Beta training on C3-Fz 2. SMR training on C4-Pz;15–22 sessions.At least a 25% increase in within sessional beta or SMR power in the 1st session.Improvement in go/no-go response time and go/no-go SD.Children showed improvements in symptoms of ADHD.Doren et al., 2017 [76] *n* = 22 (1 female) with ADHD;age range = 10–15 years, mean age = 13.4 years.Exp. = 22,no control group.Theta/beta power training was provided based on the signal at Cz,NF session—2 NF phases: 1. Puzzle task and 2. Attention task,one acquaintance session, and two theta/beta NF sessions with pre- and post-NF behavioral and EEG assessments Reduction in the theta/beta ratio and theta power during reading.Improved reading ability.No long-term measures.Escolano etal., 2014 [78]*n* = 20 with ADHD (1 female);age range = 9–13 years,mean age = 11.8 years.Exp. = 20,no control group.Increase upper alpha power averaged over six feedback electrodes (frontocentral) at IAF; 18 sessions, 5 trails per session.Enhanced upper alpha.Average increase of 13% in upper alpha power in task-related activity.Improvement in working memory, concentration, and impulsivity.A significant positive learning and improved cognitive performance over sessions.Rajabi et al., 2019 [79]*n* = 32 (32 male) with ADHD; randomized double-blind trial with two groups;age range = 6–11 years, mean age = 10.25 years.Exp. group = 16,control = 161. Beta training on FCz electrode (15 min).2. SMR training on C1–C5 (15 min).Increased activity of beta at Cz and SMR at Cz and FCz and decreased theta/beta at Cz.NF with computer training resulted in significant improvements in control of motor behavior and inhibition of attention to disturbing stimuli. Children demonstrated improvements in symptoms of ADHDChristiansen et al., 2014 [80]*n* = 58 (10 female)with ADHD; age range = 7–11 years,mean age = 8.42 years.Exp. = 32 *randomly divided into two groups;exp. group was given SCP NF training only; the control group was given medication onlySCP NF protocol;30 sessions of NF training with 40 trials of 8 min each.Positive SCPs indicated a relaxed state and negative SCPs indicated an attentive statePsychopathology ratings increased in children who did a follow-up, improved ADHD symptoms.Psychopathology ratings showed no differences from a period post-treatment to 6 months after treatment.Okumura et al., 2019 [81]*n* = 22 with ADHD; age range = 7.83–16.25 years,mean age = 12.11 yearsParticipants were divided into two groups, learners and non-learners, based on their pretraining indices and SCP regulation in training using decision tree analysis.SCP NF protocol; 10 sessions (2 sessions/day),60 trials/session;SCP recorded at Cz.Enhancement of positive SCP in learners only. No significant changes in the symptoms of ADHD.SCP might not be an effective protocol for all ADHD children.Janssen et al., 2017 [77]*n* = 38 (9 female) with ADHD; age range = 7–13 years,mean age = 9.87 years. Exp. = 38,no control group.Theta/beta NF protocol, 29 sessions (3 sessions/week),each session = 45 min,10 trials/session;inhibit theta and reinforce beta at Cz.A linear decrease in theta/beta index over sessions.No behavioral changes. Learning improved with an increase in beta power over sessions. Strehl et al., 2006 [82]*n* = 23 (9 female) with ADHD; age range = 8–13 years, mean age = not provided.Exp. = 23;no control group.SCP NF protocol; 30 sessions (3 blocks of 10) Each session = 45 min,39 trials/run, 3–5 runs/session;SCP recorded at Cz.Follow-up after 6 months.The good performance is indicated by differences in the mean amplitude of SCPs. Improvement in behavior, attention, and IQ following NF sessions.Clinical improvement is seen only in good learners. Drechsler et al., 2007 [83]*n* = 30 (7 female) with ADHD; age range = 9–13 years,mean age = 10.85 years. Randomization of group assignment was incomplete;two groups: NFT group = 17,control group = 13.SCP NF protocol; 30 sessions; 2 sessions = 45 min, 40 trials/run;SCP recorded at Cz.The good performance is indicated by differences in the mean amplitude of SCPs.Improvement in positive behavioral effects following NF sessions. Clinical improvement in ADHD symptoms is seen only in good learners. Leins et al., 2007 [84]*n* = 38 (6 female) with ADHD and were blind to group assignment; age range = 8–13 years,mean age = 9.16 years.Two groups:theta/beta NF group = 19;SCP NF group = 19. 1. Theta/beta group: theta/beta NF protocol; 30 sessions, each session = 1 hr;theta/beta recorded at C3f, C4f.2. SCP group: SCP NF protocol;30 sessions; each session = 1 hr;SCP recorded at Cz.Both groups demonstrated EEG-based learning by improvements in respective cortical activity.Both groups demonstrated improvements in attention and IQ.Clinical effects for both groups remained stable six months after treatment. Bakhshayesh et al., 2011 [75]*n* = 35 (9 female) with ADHD and were blind to group assignment; age range = 6–14 years,mean age = 9.34 years.Two groups:NF group = 18,BF group = 17.Theta/beta NF protocol, 30 sessions, each session = 30 min, 2–3 sessions/week, theta/beta recorded at Cz. Reduced theta/beta ratios in the NF group.Improved reaction times and attention following NF sessions.Improvement in ADHD symptoms was seen by parents only in the NF group. DeBeus et al., 2011 [85]*n* = 42 (29 female) with ADHD and were double blind to randomization;age range = 7–11 years, mean age = 8.8 years.Two groups:NF group = 18,placebo group = 17.NF group: suppress theta + alpha, enhance beta including SMR at Fz;20 sessions.Engagement index (beta/(theta + alpha) increase) improved following NF sessions in 74% of the children.Behavioral symptoms rated by teachers; improvements in the continuous performance test.Improvements rated by teachers correlated with the engagement index.Gevensleben et al., 2014 [86] *n* = 10 with ADHD; randomized controlled trials; age range = 10–13 years,mean age = 11.4 years.Exp. = 10;no control group.SCP NF protocol; 13 double sessions,each session = 105 min,36–38 trials/session;SCP recorded at Cz.Mean amplitude increase in SCP negativity in all trials. Decrease in inattention symptoms, and an association between mean amplitude and SCP negativity.No long-term changes.Takahashi et al., 2014 [87]*n* = 10 (3 female) with ADHD; age range = 8.4–16.6 years,mean age = 12.5 years.Exp. = 10;no control group.SCP NF protocol; 16 sessions; 2 sessions/week,each session = 12 min,60 trials/session;SCP recorded at Cz.Positive shift increase in amplitude in sessions 9 and 13, and negative shift increase in amplitude in sessions 11 and 12.No behavioral changes following NF sessions.No long-term changesNote: Exp: experimental group; IAF: individual alpha frequency, SCP: slow cortical potential. * The study was not completed by the authors, and therefore only preliminary results are provided. Fifty-eight children were part of the initial sample, but the follow-up after six months could be conducted for only 32 children. These results are the preliminary results for only the 32 children.

The second protocol is the enhancement of SMR, which helps to reduce hyper-motoric symptoms [88]. This protocol is commonly used to increase focus and attention by reducing mind-wandering and drowsiness. The enhancement of SMR in an NF context was studied by Rajabi et al. [79], who demonstrated its effectiveness in assisting children with ADHD (*n* = 32; mean age = 10.20 years). They focused on improving visual attention, concentration, and the regulation of body movements. They recorded QEEG from the participants, and the NF training was conducted at electrode sites Cz and C1–C5 (left and right central electrodes) for the first 15 sessions and at FCz and C1–C5 for the next 15 sessions under an open-eyes condition. The protocol used was to decrease theta and increase beta at FCz. In addition, SMR training was used when the children were in a hyperactive or impulsive mood to reduce theta and high beta and increase SMR at sites C1–C5. This study found medium to large effect sizes in ADHD symptoms and visual attention and a medium effect size in visual response control. DeBeus and Kaiser [85] used an engagement index (increase in beta/(theta + alpha)) in their study on children with ADHD symptoms (*n* = 42; mean age = 8.8 years). They suppressed the theta and alpha frequencies and enhanced the beta frequencies using SMR training at Fz over 20 sessions. They found a linear increase in the engagement index as the sessions progressed. The teachers rated an improvement in ADHD symptoms, which further correlated with the engagement index. This study found small to medium effect sizes for hyperactivity and impulsivity.

The third protocol involves the regulation of the slow cortical potentials (SCPs) of cortical excitations [80]. This protocol targets unexpected negative amplitude variations [89] and measures EEG signals at the vertex from several hundred milliseconds to several seconds. These signals are largely related to excitability levels in the cortical regions [90] and negative shifts of amplitudes in SCPs represent an increased excitation state, whereas positive shifts of amplitude in SCPs represent a decreased excitation state in the respective cortical areas. During SCP training, participants are typically asked to switch from a positive amplitude state (relaxed) to a negative amplitude state (attentive) or vice versa.

Christiansen et al. [80] conducted a study (*n* = 58; age = 7–11 years) to assist children with ADHD. Theirs was the first randomized controlled trial to test the effectiveness of SCPs. The children were divided into two groups: one group was administered only medication, whereas the other was administered NF. SCP NF training was incorporated to generate positive SCPs (relaxed state) and negative SCPs (attentive state). The children received 30 sessions of SCP NF training with each session consisting of 40 trials lasting 8 min each. The group without medication and with NF demonstrated greater improvements in ADHD symptoms than the group with medication only. Strehl et al. [82] used mean amplitude differences in the SCP and found that 30 sessions of 45 min each led to improvements in behavior, attention, and IQ in children with good learning ability who had ADHD symptoms (*n* = 23; age = 8–13 years). Their SCP NF training was conducted at the Cz electrode. A follow-up after 6 months indicated clinical improvements among the participants. A similar protocol-based study by Drechsler et al. [83] in children with ADHD symptoms (*n* = 30; age = 9–13 years) found similar positive behavioral improvements among those with good learning ability over 30 sessions. This study found small to medium effect sizes for hyperactivity and medium to large effect sizes for impulsivity and inattention. Okumura et al. [81] also conducted a study using SCPs and classified children with ADHD (*n* = 22; mean age = 12.11 years) as learners and non-learners. SCPs were recorded at the Cz electrode and NIRS was used to examine activations in the prefrontal cortex. De-oxygenated hemoglobin concentrations were measured from the right and left prefrontal cortex. The children who were successful in the trials and demonstrated positive SCPs were categorized as learners, whereas non-learners did not demonstrate an improvement. Leins et al. [84] compared the SCP and the theta/beta protocol for children with ADHD symptoms (*n* = 38; mean age = 9.16 years). The children were divided randomly into two groups. One group was trained with the theta/beta protocol at C3f and C4f, and the other was trained with the SCP protocol at Cz, for 30 sessions each. Both groups demonstrated improved attention and IQ following the NF sessions and the clinical outcomes were stable after 6 months of treatment, indicating successful NF training. This study found medium effect sizes for both behavioral and attention changes.

Although NF is currently used for treating or assisting individuals with ADHD symptoms, meta-analyses and a few studies on its efficacy have been inconclusive. The clinical effectiveness of NF in treating ADHD has been debated [2,91,92]. A meta-analysis by Cortese et al. [92] showed that evidence-based controlled trials with blinded outcomes are not indicative of the effectiveness of NF for treating children with ADHD [93]. Furthermore, although certain studies have mentioned NF as being “specific and efficacious,” it fails to match up to this purported efficacy in actually treating ADHD [2,91]. The inconsistent and conflicting results indicate that further studies based on high-quality controlled trials are necessary not only to validate the utility of EEG-NF and learning protocols but also to establish suitable predictors for the treatment of ADHD.

### 3.3. EEG–Neurofeedback for Assisting Children with Other Specific Learning Disorders

In addition to dyslexia and ADHD, other disorders affect learning, such as dyscalculia and dysgraphia [94]. Specific learning disorders cause difficulties in one or more areas of learning, without affecting the children’s general intelligence and motivation.

Dyscalculia is a specific neurodevelopmental learning disorder affecting numerical skills, which persists into late adolescence [95]. It is characterized by difficulties in processing numerical information, learning arithmetic facts, and performing calculations. Children with dyscalculia have comorbid difficulties in math reasoning or word reasoning accuracy. The prevalence of dyscalculia has been reported to be 3–6% in most countries, which is similar to that of ADHD and dyslexia [96]. Studies on children with dyscalculia have found deficits in the prefrontal and parietal cortices, which are involved in the performance of arithmetic tasks [97,98]. One study found a structural abnormality in the left parieto-temporal area in a patient with dyscalculia [99]. Number processing involves the intraparietal sulcus (IPS) [100], and patients with deficits in the IPS fail to perform numerical approximation and numerical comparison tasks. Dysgraphia, by contrast, pertains to deficits in handwriting performance in children [94], with a prevalence varying from 5% to 25% in adolescents [101]. A study by Richards et al. [102] compared the performance of those with good and poor writing in an fMRI finger-tapping study and found lower activations in the motor regions among those with poor writing.

Children with specific learning disorders exhibit slow EEG waves, including elevations in the theta wave and depressions in the alpha wave. They face challenges in the proper processing of information, which affects their learning, reading, and writing. This consequently affects their daily and routine functions and causes their academic achievement to be below average. Therefore, an NF system that can reduce the theta/alpha ratio in their EEG waves could be key to assisting them. Only a few studies have been conducted on children with other specific learning disorders.

Fernández et al. [47,103] characterized children with specific learning disorders as having delayed EEG with an abnormally high theta/alpha ratio. They adopted a protocol to control and reduce the abnormal theta/alpha ratio using an auditory stimulus. Children with specific learning disorders exhibited improvements in alpha and beta activity in the frontal regions indicating attention and also learned to decrease the theta/alpha ratio during the NF training. The study also found changes in EEG sources that reflected improvements in learning. Their research can serve as a baseline for all upcoming studies on the role of NF in improving specific learning disorders in children. Becerra et al. (2006) demonstrated in a follow-up study on children with learning disabilities that neurofeedback is an effective treatment for learning disorders and has beneficial effects not only immediately after neurofeedback therapy but also over a longer period [46]. In terms of cognition, the patients’ symptoms of learning disabilities improved, which is consistent with the findings of this study. Jacobs (2006) demonstrated that NF improved reading skills in two adolescents with learning disabilities. They found significant improvements in those who received the treatment when compared to the control group, as well as a lower rate of early treatment dropout, and they also reported positive results after 40 or more sessions [104]. A few uncontrolled studies on EEG-NF training that improved English reading ability have also been conducted [105]. Table 4 summarizes the results of some case studies on specific learning disorders and NF.
brainsci-12-01238-t004_Table 4Table 4Summary of studies on neurofeedback and specific learning disorders other than dyslexia.Author(s)Details of the SampleConditionsNF DetailsPost-NFBecerra et al. (2006) [46]*n* = 10 (2 female) children with learning disability; age range = 7–11 years, mean age = 11.65 years.Children diagnosed with learning disorders were divided into two groups:experimental group (*n* = 5, age = 11.2 ± 1.4, 1 female) and control group (*n* = 4 *, age = 12.1 ± 1.6, 1 female).Each child received 20 NF sessions in the experimental group, with each session lasting 30 min, and 2 sessions per week over a period of 10–12 weeks.This follow-up study lasted for 2 years with two groups: the experimental group receiving NF sessions and the control group receiving placebo treatment; verbal scores decreased after NF sessions; EEG maturational lag in control group children increased, reaching abnormally high theta values. By contrast, children in the experimental group exhibited positive behavioral changes.Jacobs (2006) Case I [104]15-year-old boy.Diagnosed with ADHD, learning disabilities in writing, reading, and spelling, and bipolar and developmental disorders.Received 40 NF sessions including right and intra-hemispheric training.Improvements in some learning deficits. Symptoms of anxiety, depression, phobias, interpersonal sensitivity, etc. improved significantly; improved focus.Jacobs (2006) Case II [104]10-year-old boy.Serious deficits in social interactions, attention, and anxiety affect his home and school functions.Received 39 NF sessions including right-hemispheric training.Improvement in inhibition and executive functions. Improved attention, social acceptability, social interaction, and control over anger.Thornton and Carmody (2005), Case II [105]9-year-old girl.History of learning problems. No academic records or neuropsychological testing was completed to verify the severity of the learning disability. Did not exhibit a high theta/low beta pattern as with other children with a learning disability.40 sessions including alpha coherence (input stage) and alpha and beta coherence (recall period).Post-NF sessions showed improvement in auditory and reading memory.Thornton and Carmody (2005), Case III [105]Boy (age not mentioned).History of reading problems. He exhibited abnormalities in connectivity and coherence.25 NF training sessions at occipital positions indicating issues in the posterior regions.Increased auditory memory functioning following the 25 sessions. In addition demonstrated improved reading scores.Thornton and Carmody (2005), Case IV [105]17-year-old.Issues relating to reading disability.20 NF sessions were provided to the participant.Following the 20 NF sessions, the comprehension scores improved from 45% to 90% (8th grade level) and 20% to 70% (10th grade level); the story recall performance score also increased.Fernández et al. [47,103]*n* = 16 with LDrandomly assigned to two groups:experimental group (*n* = 11; 6 females; age range = 7–11 years; mean age = 8.94 years): received NF training.Control group (*n* = 5; age range = 7–11 years; mean age = 9.7 years): received placebo treatment.Children with learning disability.Experimental group: Before NF training, 2–3 EEG recordings were taken for every child. Every child received 20 NF sessions (30 mins/session) for 10–12 weeks (2 sessions/week); theta/alpha ratio was calculated at the beginning and end of every session.Control group: in similar conditions, only the tone onset and its duration were randomly assigned.All children learned to decrease the theta/alpha ratio during the NF sessions. Post NF sessions, the control group did not find significant reductions in the EEG power bands. In contrast, the experimental group reduced delta and theta power levels and increased the alpha and beta power levels.Fernández et al. [47,103]*n* = 20 with LDrandomly assigned to two groups:auditory group (*n* = 10; mean age = 9.10 years): NF training using an auditory stimulus.Visual group (*n* = 10; mean age = 9.08 years): NF training using a visual stimulus.Children with learning disability.Before NF training, 2–3 EEG recordings were taken for every child. Every child received 20 NF sessions (30 mins/session) for 10–12 weeks (2 sessions/week).After the NF training, both groups significantly reduced the z-score of theta/alpha quotient; However, more children with normalized z-score theta/alpha quotient were found in the NF enforced auditory group.* One child who was included in the control group left the school one year before the completion of the study and declined to participate further in the study.

## 4. Discussion

### 4.1. Potential for Using EEG-NF in Education

As discussed in the previous sections, there are a fair number of clinical applications based on NF for the treatment of dyslexia, ADHD, and other specific learning disorders. Studies suggest that these applications are not detrimental. We identified 21 published NF group studies on children with symptoms of dyslexia, ADHD, and other specific learning disorders such as dysgraphia and dyscalculia (refer to Table 2, Table 3 and Table 4). However, although studies have demonstrated that NF can improve memory, ability to focus, and other cognitive abilities, there is little or no evidence supporting the use of NF in formal educational settings. Nevertheless, translational applications of NF in education do show some potential. These are discussed in this section.

Two studies were found on EEG-NF for children with dyslexia [45,60], both of which were based on frequency bands. They adopted protocols to strengthen various features such as power (absolute or relative) and coherence z-scores in the EEG frequency bands. These studies showed increased SMR along with increases in spelling ability and attentional modulation. The ADHD-based NF studies examined in this review adopted NF training of specific frequency bands and primarily used central electrodes [75,76,77,84]. The frequency band-based NF provided continuous feedback to the participants, targeting a particular band or a ratio of specific bands that varied across conditions. One major protocol-based study was aimed at decreasing theta activity and increasing beta activity [75]. Another characteristic protocol aimed to enhance the SMR band [34,79,85], which plays a major role in motor excitability, as suggested by previous studies [106,107]. Finally, a third protocol that was adopted in certain studies was the self-regulation of the fluctuations of the SCPs [80,81,82,83,84,86,87]. Some of these studies showed improvements in EEG regulation [75,78,86]. Other studies either reported the rate of learning [34,85] or differentiated between good and poor learners [81,82,83]. Furthermore, the six studies that we found on other specific learning disorders adopted different protocols with different sample sizes and showed inconsistent results (refer to Table 3). These case studies showed improvements in factors such as reading ability, recall ability, and attentional and working memory. However, their sample sizes were very small, making it difficult to draw conclusions from them.

A key process in biofeedback systems is the self-learning of participants with the assistance of mathematical models that evaluate the changes from the NF signals [108]. Learning can be implicit or explicit [109]. Implicit learning implies that the experimental conditions are modulated based on the learning outcomes, but this is not explicitly known to the participants. Instead, participants are notified by the neuro/biofeedback systems about the cognitive functions that need to be regulated. The role of explicit feedback in improving mathematical or reading abilities has been examined in various studies [110]. Research also indicated that children with ADHD symptoms might learn better with explicit guidance [111]. Moreover, Heinrich et al. [90] suggested that the trainer should encourage the participant to fully engage with the NF system at the beginning of the training session, and gradually reduce this encouragement when self-regulation becomes possible.

Recent research has also suggested the use of behavioral rating scales before initiating NF learning sessions to capture individual differences in learning performance [112]. Studies have also identified biomarkers of learning during an experimental NF session [113]. These results can provide informative predictors of successful learning for an educational system. NF may be used in education to facilitate learning in multiple ways. For example, a student’s emotional states during and after a class can be assessed with EEG parameters. Additionally, NF can be used to assist in focusing exercises. Improving focus and concentration will eventually result in better learning outcomes for students.

An important question is whether EEG-based NF is useful as a standalone treatment, or must it be used in conjunction with pharmacological or other methods. A few studies have adopted a between-subjects design by comparing NF as a standalone, non-pharmacological intervention in one condition with medication in the other condition [114,115,116], whereas a few EEG-based NF studies have also combined other treatment modalities [117,118]. Studies have shown that a combination of NF and other treatment modalities produces superior outcomes compared with a standalone NF system [118,119,120,121]. In summary, previous studies from both standalone and multimodal NF interventions suggest that NF as a multimodal intervention is associated with better outcomes than a standalone NF intervention. However, the actual neurophysiological mechanisms underlying the interventions remain to be tested.

Further work is also required to identify the type of interface that would be conducive to participants’ learning. This interface should not discourage the participant from performing the NF session. Rewards could play a major role in encouraging participants to complete additional sessions. The aesthetics of the interface and the electrodes are also important for educational research. These factors must be considered when developing an interface for any NF system used for education. Despite the challenges involved in the adoption of NF, numerous cognitive devices have been developed to improve attention in people affected by ADHD. NASA’s Langley research center developed a simulator to monitor a pilot’s EEGs and predict the pilot’s attention during their flight [62]. The work was further extended to the development of a video game in which the responsiveness of the joystick increases if the player produces more beta waves and fewer theta waves, and decreases if the player produces fewer beta waves and more theta waves. This resulted in the development of a player attention system with a head-mounted sensor for focusing on the game, instead of a using a joystick or any other device that requires hand–eye coordination.

### 4.2. Challenges in the Adoption of EEG-NF in Education

EEG is the most commonly used method in NF because of its high temporal resolution and non-invasiveness. However, the EEG frequency oscillations associated with a particular cognitive function, such as the understanding of a new concept, are complex and larger in scope than the attention process. Currently, EEG-NF can be used for monitoring whether students are focused during a lecture or as an assessment technique. However, to optimize learning outcomes, the system must capture meaningful neural oscillations that indicate that a certain concept has been mastered. Another issue with NF is reliability, as artifacts caused by the movement of the head or body parts during a particular task must be considered. This might also cause it to become tedious to use in an educational environment with students. This issue might be solved by the use of proper filters, feature extraction methods, and machine learning algorithms. Furthermore, the system should also be personalized to each student’s needs. To be meaningful for use in education, the NF system must be reliable, but its utility can be further enhanced if it can be personalized.

Cost is another concern for the use of NF systems in education, especially because of the need for personalization. Accurate EEG systems are also expensive and it might not be cost-effective to use them in every class. Accurate wearable cognitive devices could be a potential solution to this problem. If there are no neural markers for a particular application, event-related potentials (ERPs) may also be considered as an option [108]. NF systems must also be calibrated each time for every individual, thereby requiring additional effort to adopt. These issues outlined above should be considered during the development and implementation of educational NF systems.

### 4.3. Future Directions for EEG-NF Applications in Education

Recent studies have incorporated virtual reality (VR) in NF to understand and improve learner characteristics such as attention by examining frontal alpha oscillations [122]. Studies based on relaxation training [123], the improvement of cognitive processing [124], cognitive performance [125], positive mood states [126], and motivation [127] have been successful in training individuals using NF systems with specific protocols. A VR-based NF system can be developed in which classroom-based learning is combined with VR-related tasks and rewards to enhance the aforementioned parameters in students. This could be achieved with ease by using very large-scale integrated chips to reduce the size of the system to that of a wearable device that would help to regulate brain activity and perform tasks to achieve rewards. Ease of use must be a priority for NF to be successful in educational applications. Such NF devices would mark an advancement in the acquisition of EEGs in terms of hardware. They would also help us better understand the range of states that can be detected in an EEG. There are a few devices currently that support meditation, attention, sleep, and relaxation. Although very few of them focus on the domain of this study, we can expect more in the future. If such technologies are developed, they should be used rather than prohibited in classrooms. Children, teachers, parents, and academics should be involved to make this technology usable and useful.

The mechanism of neurofeedback is still unknown and it is still debatable whether it genuinely affects EEG or only has a placebo effect [128]. The majority of the studies included in this review demonstrated improvements in learning following NF sessions, but most studies omit the electrophysiological assessment before the treatment. To ensure that the electrophysiological changes after treatment were produced by NF, correlations of EEG before and after NF sessions are essential in the future studies.

## 5. Conclusions

There is a need to understand how children with specific learning disorders can be assisted using NF. This review reveals that there is a paucity of studies on NF that focus on improving learning abilities because research in this field is still at a nascent stage. The review also reveals that there are issues of reliability concerning the use of NF in different fields. There are arguments in the literature both against and in favor of fostering EEG-based NF learning in children with dyslexia, ADHD, and other specific learning disorders. Despite this ambivalence toward the use of NF for children with learning disorders, there has been rapid progress in the field, especially in the application of NF for education. NF can be useful for children with difficulties in numeracy and literacy, including problems with comprehension. However, further research is required before clear conclusions can be drawn. The adoption of this technology in education can lead to the development of new interactive designs and new NF protocols. In the future, NF technologies will be cost-effective, support better user interactions, and be more comfortable to use.

## Figures and Tables

**Figure 1 brainsci-12-01238-f001:**
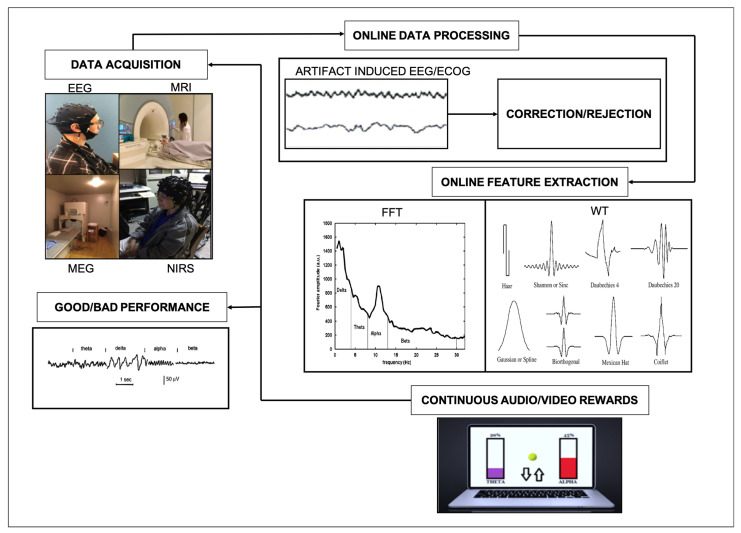
Components of an NF system. EEG: electroencephalography; ECoG: electrocorticography; MEG: magnetoencephalography; NIRS: near-infrared spectroscopy; FFT: fast Fourier transform, WT: wavelet transform.

**Figure 2 brainsci-12-01238-f002:**
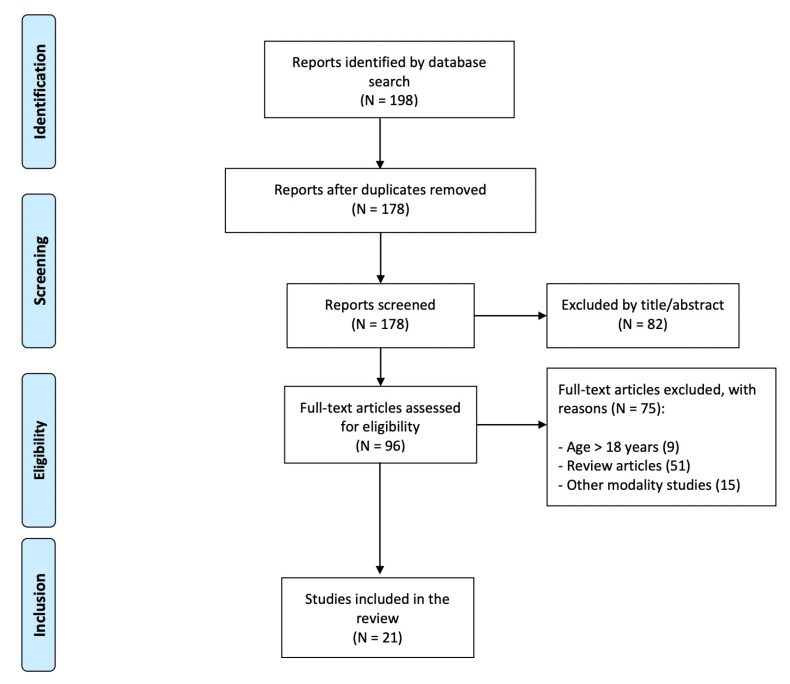
Literature search procedure.

**Table 1 brainsci-12-01238-t001:** Characteristics of EEG frequency bands.

EEg Waves	Frequency Range		Band Characteristics	
Delta	0.5–4 Hz		Unconsciousness, deep sleep, complex problem solving	
Theta	4–8 Hz		Anxiety, creativity, depression	
Alpha	Lower	8–10 Hz	Recall	Relaxation, alertness
	Upper	10–12 Hz	Cognition tasks	Peacefulness/calmness
Sensorimotor rhythm (SMR)	12–16 Hz		Relaxation, alertness	
Beta	Lower	16–20 Hz	Focus, coherent thinking, attention	
	Upper	20–30 Hz	Anxiety, attention (focused)	
Gamma	30–100 Hz		Learning, task solving	

## Data Availability

Not applicable.

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
