# Peer review of "Neurofeedback for the Education of Children with ADHD and Specific Learning Disorders: A Review"

_brainsci, 2022, doi:10.3390/brainsci12091238_

Round 1

Reviewer 1 Report

Thank you for the opportunity to review your manuscript. I see great potential in the current review.

1. I would ask the authors to clearly outline the purpose of the review and state the research questions that this review answers.

2. In addition, I would ask that the Authors make sure that literature references are reported correctly, as required by MDPI.

3. It is unclear how many papers were ultimately accepted for review, as the final roster at the end does not match the numerical counts reported in the paper's methodology.

Regards!

Author Response

1. Thank you very much for the suggestions. We made significant changes to the manuscript to provide a straightforward approach to the review's purpose.

“The review discusses the potential for translational application of NF and how it could be useful in education. In this review article, we first present a summary of a generalized NF system and then discuss the use of NF in the treatment of dyslexia, ADHD, and other specific learning disorders. Then, we make recommendations for further research and potential applications of NF in educational settings. NF-based techniques are used to treat dyslexia, ADHD, and other particular learning disorders including dysgraphia and dyscalculia. To assess the potential of NF for educational settings, the review suggests first understanding the current type of feedback provided to children, as well as the potential, challenges and limitations of the current systems, and provides future directions to aid in identifying the issues undermining the efficacy of current systems and identifying solutions to address them.”

2. We have now reported the references in accordance with MDPI's requirements.

3. Thank you for the comment. This review included twenty-one studies that met the required criteria and were included in our manuscript. The review includes two NF studies for training children with dyslexia, fourteen NF studies for training children with ADHD, and five NF studies for training children with other specific learning disorders like dysgraphia and dyscalculia (refer to Tables 2, 3 & 4).

Table 4 contains only 5 studies, although we can see a few case studies included in the same study, so the table has been referenced for greater clarity.

Reviewer 2 Report

Abstract: Please complete your abstract, this must reflect all the work that you did.

Results: The tables are great, but the font is so big. Please make this smaller.

Discussion: It is essential to highlight whether or not previous NF studies performed an electrophysiological assessment before the treatment, given that most of them omit this step. The researchers working on this topic should guarantee that the electrophysiological changes after treatment were produced by NF. I am pretty sure that this should also be part of your discussion.

Minor: Please check the font size; it is not consistent in the manuscript.

Author Response

1. The abstract in our manuscript has now been revised.

“Neurofeedback (NF) is a type of biofeedback in which an individual’s brain activity is measured and presented to them to support self-regulation of ongoing brain oscillations and achieve specific behavioral and neurophysiological outcomes. NF training induces changes in neurophysiological circuits that are associated with behavioral changes. Recent evidence suggests that NF technique can be used to train electrical brain activity and facilitate learning in children with learning disorders or disorders that affect learning. To that end, this review first presents a generalized model for NF systems before providing literature evidence for studies involving NF training for children with disorders such as dyslexia, attention-deficit/hyperactivity disorder (ADHD), and other specific learning disorders that affect learning. The review goes into greater detail about the potential for translational applications of NF in educational and learning settings. The review also addresses some issues concerning the role of NF in education, and it concludes with some solutions and future directions. In order to provide the best learning environment for children with ADHD and other learning disorders, it is critical to better understand the role of NF in educational settings. The review provides the potential, challenges of current systems to aid in highlighting the issues undermining the efficacy of current systems and identifying solutions to address them. The review focuses on the use of NF technology in education for the development of adaptive teaching methods and the best learning environment for children with learning disabilities.”

2. We have adjusted the font size of the Tables.

3. Thank you for the suggestion. In the Discussion section, we have included a paragraph emphasizing the importance of electrophysiological assessment prior to NF sessions in NF studies.

“Since the mechanism of neurofeedback is still unknown and it is still debatable whether it genuinely affects EEG or only has a placebo effect [130]. The majority of the studies included in this review demonstrated improvement in learning following NF sessions, but most studies omit the electrophysiological assessment before the treatment. To ensure that the electrophysiological changes after treatment were produced by NF, correlations of EEG before and after NF sessions are essential in the future studies.”

4. The manuscript font size is now consistent.
